# Epigenetic Perspective of Immunotherapy for Cancers

**DOI:** 10.3390/cells12030365

**Published:** 2023-01-19

**Authors:** Sunita Keshari, Praveen Barrodia, Anand Kamal Singh

**Affiliations:** 1Department of Immunology, The University of Texas MD Anderson Cancer Center, Houston, TX 77054, USA; 2Department of Genomic Medicine, The University of Texas MD Anderson Cancer Center, Houston, TX 77054, USA

**Keywords:** cancer, immunotherapy, epigenetics, immune checkpoint drugs, epigenetic drugs

## Abstract

Immunotherapy has brought new hope for cancer patients in recent times. However, despite the promising success of immunotherapy, there is still a need to address major challenges including heterogeneity in response among patients, the reoccurrence of the disease, and iRAEs (immune-related adverse effects). The first critical step towards solving these issues is understanding the epigenomic events that play a significant role in the regulation of specific biomolecules in the context of the immune population present in the tumor immune microenvironment (TIME) during various treatments and responses. A prominent advantage of this step is that it would enable researchers to harness the reversibility of epigenetic modifications for their druggability. Therefore, we reviewed the crucial studies in which varying epigenomic events were captured with immuno-oncology set-ups. Finally, we discuss the therapeutic possibilities of their utilization for the betterment of immunotherapy in terms of diagnosis, progression, and cure for cancer patients.

## 1. Introduction

Immunotherapy (a type of cancer treatment that relies on empowering the body’s own immune cells to fight cancer) has expanded to include: i—immune checkpoint/ligand inhibitors (CTLA-4, PD-1, PD-L1/L2, TIM3, and TIGIT) [1,2,3,4,5,6,7,8,9,10,11,12]; ii—adoptive T-cell transfer therapy (CAR-T, TCR-T, TIL and NK cell) [13,14,15,16,17]; iii—cancer vaccines (T-vec, BCG and Sipuleucel-T) [18,19]; and iv—immunomodulators (thalidomide, lenalidomide and pomalidomide) [20,21,22,23]. Immunotherapy has revolutionized cancer treatment, providing significant clinical benefits to patients with different types of cancers. However, only a small subset of patients benefit from immunotherapy, which highlights limitations of this therapy. Major limitations include the low response rate evidenced by primary/acquired resistance and iRAEs [24,25,26,27]. These limitations can be attributed to epigenetic changes acquired by the TIME that play an imperative role in the development of intra/inter tumor heterogeneity by favoring the evolution of transcriptionally distinct clonal populations of cancer cells, which ultimately aid tumor progression and development [28,29].

Epigenetic aberrations are considered hallmarks of cancer development and progression [30]. In the TIME, cancer cells escape immune-mediated cell death by utilizing epigenetic mechanisms to escape host immune recognition and immunogenicity [31,32]. In the tumor microenvironment (TME), in addition to cancer cells, immune cells also undergo various epigenetic modifications that alter their effector cytokine expression, cancer immunosurveillance, immune-checkpoint molecule expression, and tumor-associated antigen presentation with MHC molecules [33,34]. Additionally, epigenetic modulators such as DNA methyltransferase inhibitors (DNMTis) and histone deacetylase inhibitors (HDACis) can re-program the TIME to increase the susceptibility of tumor cells to cytotoxic T-cell-mediated killing, leading to enhanced anti-tumor immune responses [35,36]. Moreover, unlike genetic alterations, epigenetic modifiers can be pharmacologically altered to revert the changes acquired during cancer initiation and progression [37,38,39,40].

An improved understanding of epigenetic events related to immunotherapy resistance would be helpful in designing potential combination strategies for immunotherapy. Multiple factors including constitutive PD-L1 expression in cancer cells, a lack of tumor antigens, defective antigen presentation and processing machinery, the exhaustion of infiltrated T cells, and the presence of an immunosuppressive population—such as Tregs, myeloid-derived suppressor cells (MDSCs), and tumor-associated macrophages (TAMs)—could contribute to acquired resistance to immunotherapy [41,42,43] for TAMs [44].

Therefore, combination therapies involving epigenetic drugs/targets and immunotherapy can serve as improved therapeutic strategies for cancer management by boosting anti-tumor immunity.

## 2. Epigenetic Modifiers

Epigenetic modification involves a broad range of heritable and reversible changes in gene expression without altering DNA sequences [45,46]. Epigenomic modifications regulate transcription via the modulation of chromatin through the following mechanisms: (1) the post-translational modifications (PTMs) of histone proteins, (2) CpG methylation/demethylation, (3) ATP-dependent nucleosomal repositioning, (4) histone variant exchange, and (5) the action of noncoding RNAs (such as micro RNAs) and (6) chromatin loops [45,47]. Histone tail chemical modifications such as acetylation, methylation, and DNA methylation, which are heritable marks and crucial for the accurate transmission of chromatin states and subsequent gene expression, are the most studied epigenetic modifications [48]. Importantly, these epigenetic modifications are profoundly altered in tumor generation and progression [49]. One important unanswered question is which hierarchical order of events leads to altered gene expression during cancer development. The enzymes involved in epigenetic modifications include DNA methyltransferases (DNMTs), DNA demethylases, histone methyltransferases (HMTs), histone demethylases (HDMs), histone acetyltransferases (HAT), and histone deacetylases (HDACs). DNMT inhibitors (DNMTis) and HDAC inhibitors (HDACis) are the most common epigenetic modulators in clinical use; they, along with the immune modulators, have been identified to regulate the function of immune cells in multiple tumor types (Table 1).

## 3. Epigenetic Modifiers in T Cells

The functional differentiation of T cells, like short-lived effectors, long-term memory T cells, Treg, and other T-cell populations, is majorly influenced by epigenetic modifications. An increasing number of investigations support the crucial role of HATis, HDACis and HMTs in regulating the fate and function of T cells. The inhibition of HDAC1 and HDAC2 promote the differentiation of CD4^+^ T cells into cytotoxic CD4^+^ T cells [50,51]. HDAC3 is critical for the maturation of both CD4+ and CD8+T cells and the production of TNF upon TCR/CD28 stimulation [52]. Enrichment in the central memory and stem cell memory phenotypes of T cells is regulated by H3K4me3 modification at specific gene promoters such as TCF7, LEF1, and KLF2. Interestingly, the upregulation of H3K4me3 and the downregulation of H3K27me3 at the Gcnt1 locus were found to enhance the trafficking of memory T cells to tumor sites in an interleukin (IL)-15-dependent manner [53].

Scheer et al. reported that lysine methyltransferase Dot1l-dependent H3K79me2 is crucial for CD4+ T helper (Th) cell differentiation, as the loss of it was found to lead to the increased expression of Th-1-specific genes and the overproduction of IFN-γ at the expense of Th-2 cell development, advocating a central role for Dot1l in Th-2 cell lineage commitment and stability [54]. Another study investigated the role of menin, a major component of the trithorax group (TrxG) using Cd4-cre-driven conditional knockout (KO) mice; a deficiency in menin was shown to lead to the downregulation of Gata3 expression due to reduced levels of H3K9ac and H3K4me3 at the upstream regions of the Gata3 proximal promoter [55]. Interestingly, the suppression of histone H3K27 demethylases KDM6A (UTX) in mature Th-17 cells was found to reduce mitochondrial biogenesis, causing metabolic reprogramming and reducing the expression of key metabolic TFs, such as PPRC1, which ultimately showed anti-inflammatory effects [56]. The results of these studies reinforce the role of epigenomic events in T-cell biology.

### 3.1. Epigenetic Modifiers in Immune Checkpoint Therapy

A critical balance between immune co-inhibitory and co-stimulatory signals in the TIME is maintained to restrict tumor development and progression (Figure 1) [57,58]. The epigenetically regulated aberrant expression of immune checkpoints (ICs), including PD-1, CTLA-4, TIM-3 (T-cell immunoglobulin and mucin-domain containing-3), LAG-3 (lymphocyte-activation gene 3), TIGIT (T-cell immunoreceptor with Ig and ITIM domains), VISTA (V-domain Ig suppressor of T-cell activation), CD276 (B7-H3), B7-H4 (VTCN1/B7x/B7S1/B7 homolog 40), IDO-1 (indoleamine 2,3-dioxygenase 1), CD161, CD38, CD93, and CD47 may result in the induction of an immune-suppressive environment, which helps tumor cells to evade immune destruction [12,59,60]. Targeting altered epigenetic modifications can significantly contribute to the reversal of the transcriptomic regulation of ICs and their ligands, which could help to re-establish potent host immunosurveillance mechanisms [61].

DNMTis and HDACis have been shown to cause the upregulation of immune-signaling components and antigen presentation through the expression of ERVs (endogenous retroviral sequences), thereby improving tumor cell recognition [62]. Decitabine (DNMTi) upregulates the cancer testis antigen member *MAGE-1* via hypomethylation, thus increasing chances of its presentation through MHC molecules to effector immune cells [63]. Panobinostat (pan HDACi) has been found to significantly increase CD38 expression in multiple myeloma, so it has been utilized in the development of an effective combinatorial treatment with daratumumab [64]. Panobinostat also affects the PD-L1/PD1 axis via the upregulation of PD-L1 in melanoma cells, which can be then targeted with anti PD-L1 antibodies [65].

H3K9 lysine methyltransferase, SETDB1, has a critical role in the carcinogenesis of multiple tissue types through the transcriptional silencing of multiple genes at specific loci. The amplification and increased expression of SETDB1 in advanced clear renal cell carcinoma has been found to be associated with a poor response to anti-PD1 therapy [66]. Further exploring this information, the Bernstein lab identified that the reversal of the epigenetic silencing of *SETDB1* activates tumor immunogenicity through the hypomethylation of H3K9 in the transposable elements that reside in the MHC peptidome [67,68]. The results of these studies indicate the high potential of SETDB1 inhibitors such as mithramycin in combination with immune checkpoint blockade therapy (ICT) [69].

The H3K27Ac reader bromodomain and extra-terminal motif (BET) protein is overexpressed in various cancers and involved in the regulation of the (PD-1/PD-L1) immune checkpoint axis [36]. Accordingly, targeting BET with JQ1 inhibitors in combination with anti-PD1 therapy has been proven to be effective in ovarian and triple-negative breast cancer [36,70]. In another study, Adeegbe et al. showed that JQ1 treatment significantly lowered PD-L1 expression in tumor cells, which led to an increased tumor infiltration of cytotoxic T cells in a non-small cell lung cancer NSCLC xenograft, and a combination treatment of JQ1 with anti-PD-1 reduced tumor burden and resulted in an improved survival rate [71].

The promoter hypomethylation of LAG3 has been a major epigenetic regulator of mRNA expression in clear cell renal cell carcinoma (KIRC), which has a proven association with increased immune cell infiltration and an interferon-γ signature [72]. Beyond cancer, patients’ aberrant histone methylation in chronic osteomyelitis is related to the higher expression of LAG3 in the T cells of peripheral blood [73]. Another negative stimulatory molecule, Tim-3, has been shown to be epigenetically regulated, so its increased expression inhibits the expansion of Th1 and Th17 responses via its binding to galectin-9, ultimately leading to immune exhaustion in the tumor microenvironment [74,75,76]. EZH2-H3K27me3/DNMT3A-DNA methylation regulates the expression of Tim-3 and galectin-9 in HPV18-associated cervical cancer [77]. Tim-3 and galectin-9 are overexpressed in cervical cancer cases, which is mediated through the hypomethylation of *HAVCR2* and *LGALS9* because of the lesser expression and recruitment of DNMT3A to their promoter regions. SUV39H1, a H3K9me3-specific histone methyltransferase, contributes to Tim-3 and galectin-9 regulation by upregulating the H3K9me3 level at the DNMT3A promoter region, hence downregulating its expression. Therefore, SUV39H1 can be utilized as a potential therapeutic target that can downregulate the immune checkpoint inhibitors Tim-3 and galectin-9 [78].

Another immune checkpoint, TIGIT, was found to be upregulated during T-cell and NK-cell exhaustion [9]. Moreover, TIGIT was reported to be regulated by promoter demethylation in melanoma, thus making it sensitive to anti PD-1 therapy [79].

### 3.2. Epigenetic Modifiers in Antigen Processing and Presentation

In a proper functioning immune system, T cells recognize tumor antigens based on the binding of a T-cell receptor (TCR) and a matching antigen packaged into major histocompatibility complex (MHC) proteins on APCs. Tumor cells escape immune recognition through multiple mechanisms such as alterations in antigen presentation and processing machinery (APM) or alterations in MHC class I molecules, which further impair their identification by CTLs (Figure 2).

An efficient cancer immunotherapy depends on the recognition of antigens loaded onto the MHC molecules of antigen-presenting cells by T cells in the TIME. The epigenomic regulatory factors that can influence the T-cell recognition of tumor antigens include: (1) the aberrant expression of genes involved in the processing or presentation of tumor antigens and (2) the aberrant expression of antigens. There is a subclass of cancer testis antigens (CTAs), including MAGE (melanoma-associated antigen), PRAME (preferentially expressed antigen of melanoma) and NY-ESO-1 (New York esophageal squamous cell carcinoma-1), which are controlled by DNA methylation and remain silenced in mature somatic cells but are demethylated and overexpressed in various cancers [80,81]. Guadecitabine (SGI-110) and decitabine, which are hypomethylating drugs, have been shown to upregulate/overexpress CTAs such as NY-ESO-1 in epithelial ovarian cancer cells and xenografts when used in combination with NY-ESO-1 vaccine and doxorubicin chemotherapy; T-cell responses to NY-ESO-1 have been observed in most studied patients [82,83].

Studies have evidenced that DNMTis and/or HDACis could alter the expression of MHC class I molecules in cancer cells such as neuroblastoma, cervical, and prostate cancer [84]. Furthermore, the expression of different components of the APM pathway such as TAP-1, TAP-2, LMP2, LMP7 and tapasin can be manipulated by both DNMTis and HDACis in different tumor types [85,86,87]. DNMTis and HDACis can regulate the expression of the costimulatory molecules ICAM-1, CD40, CD80, and CD86 [86,88,89].

Histone methyltransferase SETDB1, which maintains heterochromatin (H3K9me3), plays crucial roles in the carcinogenesis of multiple tissue types through the transcriptional silencing of multiple genes [90]. Accordingly, the inhibition of SETDB1 was found to enhance specific cytotoxic T-cell responses against tumors via the activation of immunostimulatory genes, the encoding of retroviral antigens, and the generation of neoantigen MHC-I peptides, thus suggesting that SETDB1 has high potential to synergize with ICT [91]. The HDAC-1/3 inhibitor entinostat, upon combination with a PD-1 axis blockade, was found to lead to the complete remission of tumors, the expansion of neoantigen-specific T cells, and the induction of long-term immunologic memory in immune-competent bladder cancer mouse models [92].

### 3.3. Epigenetic Modifiers in Tumor-Infiltrating Immunosuppressive Cells

Tumor-infiltrating immunosuppressive cells such as myeloid-derived suppressor cells (MDSCs), tumor-associated macrophages (TAMs), regulatory T cells (Tregs), and cancer-associated fibroblasts (CAFs) inhibit T cells’ effector functionality and anti-tumor responses, which lead to the immune escape of tumors. The presence of an immunosuppressive cell population in the TIME could be a major contributory factor in ineffective ICTs [93]. HDACis have antitumor effects in that they reduce the number of MDSCs through various mechanisms of action such as CG-745, a class I–IIb HDACi that induces the infiltration of lymphocytes by increased antigen presentation and that decreases the amount of MDSCs by decreasing the polarization of M2 macrophages in tumors [35]. Valproic acid (VPA), a class-I HDACi, attenuates the immunosuppressive function of MDSCs by downregulating the expression of retinoblastoma 1 (Rb1), toll-like receptor 4 (TLR4), programmed cell death 1 ligand (PD-L1), and interleukin-4 receptor-alpha (IL-4Ra)/arginase [94]. Moreover, the combinatorial treatment of VPA and anti-PD-1 antibodies was found to repress the growth of B16F10 and EL4 tumor models by impairing tumor-infiltrating M2-MDSC accumulation in the tumor microenvironment compared with their individual therapies [95]. Thus, treatment with epigenetic modifiers inhibits MDSC accumulation, thereby augmenting immune checkpoint inhibitors for successful cancer treatment. Vorinostat (suberoylanilide hydroxamic acid, SAHA), a class I–II–IV HDACi, was shown to have anti-tumor potential for a 4T1 mammary mice model in which it decreased MDSC accumulation in the spleen, blood, and tumor while promoting the activation and function of CD8+ T cells [96].

Tregs play significant roles in inducing variety of immune responses, as determined by the expression of Foxp3, a transcription factor in natural Tregs (nTregs) in the thymus [97,98]. Extrinsic molecular signals including IL-2 and TCR, along with a network of transcription factors, are critical for regulating the expression of Foxp3 through epigenomic modulation, which ultimately determines a Treg’s phenotypic plasticity [99,100]. Epigenetic modifiers such as DNMT1 and DNMT3b are differentially bound to Foxp3 promoter and enhancer sites in nTregs compared with extrinsically induced Tregs. Importantly, DNMTis demethylate and activate the Foxp3 promoter and enhancer elements to induce Foxp3 expression and subsequently enable the induction of Foxp3-dependent, Treg-restricted sets of genes [101]. Demethylation in synergy with TGF-β transforms naive T cells into Tregs with high Foxp3 expression and potent, stable suppressive function [102].

Foxp3 expression in Treg cells was found to be significantly upregulated upon treatment with trichostatin-A (TSA), a HDACi [103]. Moreover, the CTLA4, PD-1, GITR and IL-10 genes are reportedly upregulated by TSA [104]. Ohkura et al. reported that Treg maturation, Treg-specific gene expression, and Treg-specific immunosuppressive activity involve epigenetic regulation through genome-wide CpG DNA hypomethylation pattern [105]. In other study, Wang et al. showed that the inhibition of EZH2, a histone-lysine N-methyltransferase enzyme, resulted in Treg-mediated pro-inflammatory activities in the TME, supporting the idea of the generation of an effector T-cell-mediated anti-tumor immune response [106].

### 3.4. Epigenetic Modifiers in Inflammatory Cytokines and Chemokines

The pro-/anti-tumorigenic effect of inflammatory cytokines and chemokines, such as TNF-α, IL-1, IL-6, and IFN-γ, has been well-established in tumor malignancies; however there is little evidence that their aberrant expression is regulated through various epigenetic mechanisms in cancer development [107,108]. IFN-γ is a pleiotropic cytokine associated with the induction of reactions in T lymphocytes, which contributes to the enhancement of an immune response against malignant cells. The downregulation of IFN-γ mediated by hypermethylation has been observed in lung and cervical cancer [109,110]. Interestingly, IFN-γ is suppressed in the presence of E6 (a human papillomavirus (HPV) protein), suggesting the involvement of E6 in IFN-γ de novo methylation followed by transcriptional silencing [111]. One of the earliest studies in humans showed that epigenetic modifications occurring in the IFN-γ, IL-4, and IL-13 genes regulate the differentiation of CD4 T cells into Th1 and Th2 cell lineages. The IFN-γ promoter is demethylated during differentiation into Th1 cells [112], and the demethylation of several specific CpG dinucleotides occurs in the IL-4 and IL-13 genes during Th2 differentiation [113]. Most importantly, epigenetic histone marks are major determinants of Th1/Th2 cell fate.

In addition to their role in development and inflammatory responses, chemokines and their receptors also play critical roles in neoplastic transformations, cancer progression, and angiogenesis. CXCL14 (also known as BRAK), a member of the chemokine family, acts as a chemoattractant and stimulates the trafficking of natural killer cells to sites of inflammation or malignancy [114]. The aberrant methylation of CpG islands in the promoter region and the first exon of the CXCL14 gene is associated with its downregulation in gastric cancer [115]. Moreover, CXCL14 was found to be transcriptionally inactivated by promoter CpG hypermethylation in human prostate cancer [116]. CXCL12 and its receptor CXCR4 belong to same family of CXCL14 and are also associated with tumorigenesis. Interestingly, the demethylation of CXCR4 and the hypermethylation of CXCL12 and ESR1 are predictive marker of tumor stage, size, metastasis, and poor overall survival in breast cancer [117].

Multiple proinflammatory cytokines including interleukins are often stated to be epigenetically regulated in various forms of cancer, especially lung cancer. The expression of the IL-1B, IL-6, and IL-8 genes are regulated through promoter DNA methylation which have been reported to play crucial roles in lung cancer [118]. Interleukin-23, a member of the IL-6 superfamily, is stated to be epigenetically regulated in non-small-cell lung cancer (NSCLC) via both histone acetylation and DNA methylation [119]. The epigenetic silencing of IL12RB2, a subunit of the IL-12 receptor, is a recurrent event in human lung cancers [120]. Furthermore, IL12RB2 methylation has been found to be frequent in patients suffering from both chronic obstructive pulmonary diseases (COPD) and non-small-cell lung cancer (NSCLC) [121].

### 3.5. Epigenetic Modifiers in Natural Killer Cells

NK cells are key mediators of the innate immune response, and they exert cytotoxic effects after the recognition of cancer cells and virus-infected cells [122]. Upon the recognition of tumor cells, NK-cell activation occurs through the interaction of NKG2D receptors on the surface of NK cells with ULBP ligands and the MHC class I chain-associated proteins MICA and MICB on the surface of tumor cells [123]. One study reported that VPA (a HDACi) upregulates NKG2D, the immunoreceptor that binds with MICA and MICB, thus leading to the enhancement of the NK-mediated lysis of cancer cells in AML [122,124]. EZH2, an HMT, was found to inhibit the differentiation and function of NK cells by downregulating NKG2D receptor expression. Moreover, EZH2-mediated H3K27me3 induces the silencing of the IL-15R, CD122, and NKG2D receptor proteins, hence suppressing NK-cell expansion and decreasing the cytotoxic targeting of tumor cells.

### 3.6. Epigenetic Modifiers in CAR-T Therapy

Cell-based therapies, such as chimeric antigen receptor (CAR) T-cell therapies, have led to enormous successes against several hematological malignancies. However, their success has been limited due to tumor antigen heterogeneity, tumor infiltration, and persistence. Nevertheless, the manipulation and modification of epigenetic and genetic cascades have been observed to trigger specific T-cell-signaling pathways, which can help to promote the expansion and persistence of CAR-T cells. In support of this idea, a hypomorphic mutation in the epigenetic modifier TET2 (a chromatin modifier that encodes the methyl cytosine dioxygenase enzyme that facilitates DNA demethylation to activate gene expression) has been shown to support the central memory phenotype in anti-CD19 CAR-T cells in chronic lymphocytic leukemia [125]. The inhibition of TET2 through S-2-hydroxyglutarate (S-2HG) was found to result in the formation of CD8+ central memory CAR T cells, which helps to overcome the issue of the persistence of CAR-T cells in patients with B-cell malignancies [126]. The pretreatment of lymphoma cells with decitabine, a DNMTi, was shown to lead to the increased expression of the surface antigen CD19 on lymphoma cells, making them more susceptible to CD19 CAR T cells; this was observed in two lymphoma patients who were treated with decitabine before CAR T-cell therapy and achieved complete remission [127]. The triple knockdown of the T-cell exhaustion signature genes PD-1, Tim-3, and Lag-3 dramatically increases the chromatin accessibility of the CD56 gene, leading to the increased expression of CD56 in CAR-T cells and making them more effective at infiltrating ovarian cancer [128]. Antigen heterogeneity and antigen loss are key obstacles for developing effective CAR T-cell therapy. EZH2 is associated with a low antigen presentation and poor immunogenicity, and targeting EZH2 with a selective inhibitor and CAR T-cell therapy was found to lead to the significant enhancement of the antitumor activity of CAR T cells [129].

## 4. Transcription Factor Circuitry in ICT Resistance

ICT resistance can be attributed to T-cell exhaustion and a lack of central and effector memory T-cell formation. Numerous transcription factors are associated with these T-cell fates, and a few of them have been reported to be epigenetically regulated including NR4A1, NR4A2, NR4A3, TBET, EOMES, TCF1 (TCF7), LEF1, BATF, NFAT and EGR2 [130,131]. The high expression of NR4A1, NR4A2, and NR4A3 has been related to the poor prognosis of many cancers, and there are reports that suggest that their binding to the target LEF1 promoters can be regulated through DNA methylation and histone acetylation levels [132]. T-bet has the ability to recruit a H3K27-demethylase-Jmjd3 and a H3K4-methyltransferase-Set7/9 complex to its target genes that empower T-bet to effectively modulate the epigenomic state of its target genes and T-cell fate [133]. EOMES has been reported to be a member of chromatin-modulating complexes containing BRG1, which has been observed in in RUNX3 enhancers in T-cell innate memory formations [134]. BATF epigenetically regulates activation-associated gene expression in tumor-infiltrated Treg cells [135]. Networks of TFs also determine the differentiation and phenotypes of the T-cell-like T-bet, which displaces Sin3A-histone deacetylase (HDAC1 and HDAC2) complexes to facilitate the differentiation of Th1 cells [136]. In response to IL-12 signals, the activation of STAT4 (required for the development of Th1 cells), facilitates chromatin remodeling at the enhancer regions of Th1 genes. EZH2 facilitates the correct expression of Tbx21 and GATA-3 for differentiating Th1 and Th2 cells through H3K27 trimethylation (H3K27me3) [137]. BATF regulates Th1 gene expression via the acetylation of T-bet and IFN-γ, considered an important checkpoint in T-cell differentiation [138].

## 5. Conclusions

Immunotherapy is a major breakthrough in cancer treatment, though it still has challenges. If we can better understand immunotherapy at the cellular and molecular levels, then we can deal with its underlying issues. This review is a step towards understanding the epigenetic mechanism involved in the significant components of immunotherapy. This understanding can enable us to develop new ideas and hypotheses in the study of novel combinatorial treatment/biomarkers for correct treatment plans, including epigenetic features that have the intrinsic advantage of “reversibility”. This will increase the chances of success in future cancer immunotherapies.

## Figures and Tables

**Figure 1 cells-12-00365-f001:**
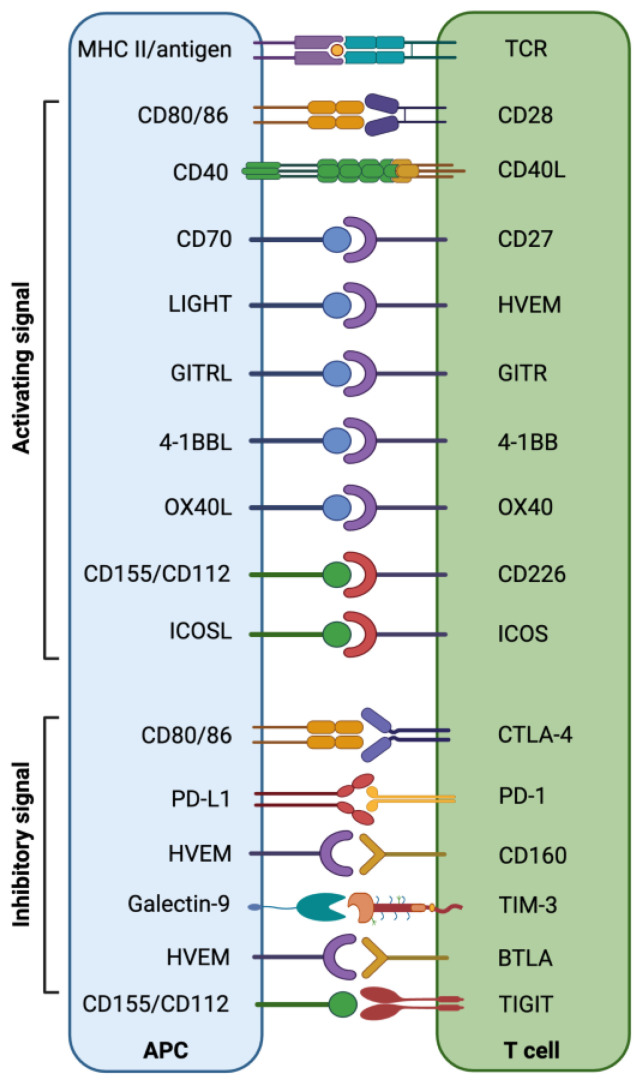
Interaction of co-stimulatory/inhibitory molecules between T cells and APCs/tumor cells provides an overview of the immune checkpoint/stimulatory molecules involved in the anti-tumor immune response.

**Figure 2 cells-12-00365-f002:**
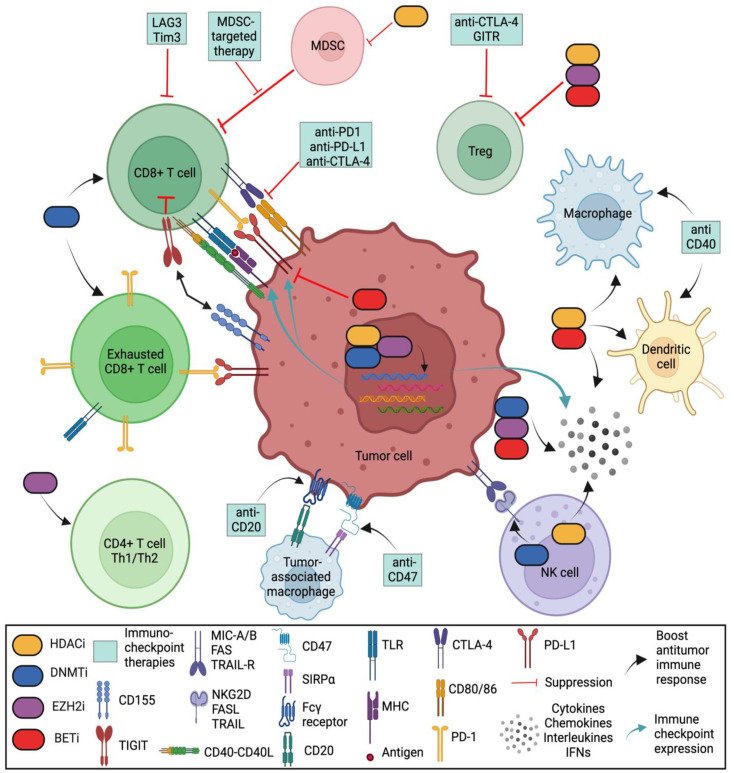
Role of various epigenetic modifiers in the tumor immune microenvironment. DNA methyltransferase inhibitors (DNMTis), histone deacetylase inhibitors (HDACis), an inhibitor of histone methylation on histone H3 at lysine 27 (EZH2i), and inhibitor of bromodomain and extra-terminal motif (BETi) shape the tumor-immune microenvironment by (i) increasing the number of CD8 and CD4 T cells; (ii) activating antigen processing and presentation machinery; (iii) decreasing the abundance of MDSCs and tumor-associated macrophages (TAMs); (iv) downregulating the immune checkpoint inhibitors Tim-3, Lag-3 and TIGIT; (v) upregulating immune checkpoint PD-L1 (by DNMTis, HDACis and EZH2i) and downregulating PD-L1 (by BETi); (vi) enhancing NK-mediated lysis (by HDACis) or decreasing NK cytotoxicity (by EZH2i); and (vii) upregulating inflammatory genes and pathways that control the secretion of interferons (IFNs), cytokines, and chemokines from tumor cells. (Regulatory T cells (Tregs) are a specialized subpopulation of T cells that act to suppress the immune response, thereby maintaining homeostasis and self-tolerance. It has been shown that Tregs are able to inhibit T-cell proliferation and cytokine production, as well as play a critical role in preventing autoimmunity. Tumor-associated macrophages (TAMs) are the key cells that create an immunosuppressive tumor microenvironment (TME) by producing cytokines, chemokines, and growth factors and by triggering the inhibitory immune checkpoint proteins release in T cells. Natural killer (NK) cells are effector lymphocytes of the innate immune system that control several types of tumors and microbial infections by limiting their spread and subsequent tissue damage. Cancer-associated fibroblasts (CAFs) are one of the most abundant and critical components of the tumor mesenchyme; they not only provide physical support for tumor cells but also play a key role in promoting and retarding tumorigenesis in a context-dependent manner. Recent studies have revealed their roles in immune evasion and poor responses to cancer immunotherapy).

**Table 1 cells-12-00365-t001:** Clinical trials of epigenetic agents combined with immune checkpoint inhibitors for cancer therapy.

NCT Identifier	Malignant Conditions	Therapeutics (Single or Combined)	Start Year	Status
NCT05089370	•Malignant Melanoma	•Combination Product: Oral Decitabine/Cedazuridine (DNMT inhibitor) in Combination with Nivolumab (PD-1 inhibitor)	2022	Recruiting
NCT04705818	•Advanced Solid Tumor •Advanced Colorectal Carcinoma •Advanced Soft tissue Sarcoma •Advanced Pancreatic Adenocarcinoma •Adult Solid Tumor	•Drug: Durvalumab (PD-L1 inhibitor) •Drug: Tazemetostat (EZH2 inhibitor)	2021	Recruiting
NCT04648826	•Sarcomas •Melanomas •Germ Cell Tumors •Epithelial Malignancies (Excluding Lung and Renal Cell Carcinomas) •Pulmonary Metastases	•Drug: Bintrafusp alfa (bifunctional fusion protein composed of the extracellular domain of the TGF-receptor II fused to an IgG1 antibody blocking PD-L1) •Drug: Azacytidine (DNMT1 inhibitor)	2021	Withdrawn
NCT04190056	•Anatomic Stage IV Breast Cancer AJCC v8 •Prognostic Stage IV Breast Cancer AJCC v8	•Biological: Pembrolizumab (PD-1 inhibitor) •Drug: Tamoxifen (antiestrogen) •Drug: Vorinostat (HDAC inhibitor)	2021	Recruiting
NCT04471974	•Castration-Resistant Prostate Carcinoma •Metastatic Prostate Adenocarcinoma •Metastatic Prostate Small Cell Carcinoma •Stage IV Prostate Cancer AJCC v8 •Stage IVA Prostate Cancer AJCC v8 •Stage IVB Prostate Cancer AJCC v8	•Drug: ZEN-3694 (BET bromodomain inhibitor) •Drug: Enzalutamide (nonsteroidal antiandrogen (NSAA) medication) •Biological: Pembrolizumab (PD-1 inhibitor)	2021	Recruiting
NCT04708470	•Cancer •Solid Tumor •Metastatic Checkpoint Refractory HPV-Associated Malignancies •Microsatellite Stable Colon Cancer (MSS)	•Drug: Bintrafusp Alfa (bifunctional fusion protein composed of the extracellular domain of the TGF-receptor II fused to an IgG1 antibody blocking PD-L1) •Drug: NHS-IL12 (tumor-targeting immunocytokine) •Drug: Entinostat (HDAC inhibitor)	2021	Recruiting
NCT04257448	•Pancreas Cancer •Pancreatic Adenocarcinoma •Pancreatic Ductal Adenocarcinoma	•Drug: Romidepsin (HDAC inhibitor) •Drug: Azacitidine (DNMT inhibitor) •Drug: Nab-Paclitaxel (stops cancer cells from separating into two new cells) •Drug: Gemcitabine (induces interferon signaling) •Drug: Durvalumab (PD-L1 inhibitor) •Drug: Lenalidomide capsule (potent molecular analog of thalidomide)	2020	Recruiting
NCT04611711	•Patients With Digestive System Tumors Resistant to PD-1 Inhibitors	•Drug: Decitabine (DNMT inhibitor) + TQB2450 injection (PD-1 inhibitor) •Drug: Decitabine (DNMT inhibitor) + TQB2450 injection (PD-1 inhibitor) + Anlotinib (VEGFR inhibitor)	2020	Not yet recruiting
NCT04553393	•Refractory or Relapsed Aggressive r/r BNHL With Huge Tumor Burden	•Drug: Chidamide (HDAC inhibitor) •Drug: Decitabine (DNMT inhibitor) •Biological: Decitabine-Primed Tandem CAR19/20-Engineered T Cells	2020	Recruiting
NCT04407741	•Solid Tumor •Lymphoma	•Drug: SHR2554 (EZH2 inhibitor) •Drug: SHR1701 (PD-1 and TGF-β inhibitor)	2020	Recruiting
NCT04414969	•Immune Checkpoint Inhibitor •Chemotherapy Effect •Epigenetic Disorder •NK/T-Cell Lymphoma of Nasal Cavity	•Drug: Anti-PD-1 antibody + Peg-Asparaginase + Chidamide (HDAC inhibitor)	2020	Recruiting
NCT04250246	•Melanoma •Non-Small Cell Lung Cancer	•Drug: Ipilimumab (CTLA-4 antibody) •Biological: Nivolumab (PD-1 inhibitor) •Drug: Guadecitabine (DNMT inhibitor)	2020	Not yet recruiting
NCT04277442	•Acute Myeloid Leukemia	•Drug: Decitabine (DNMT inhibitor) •Biological: Nivolumab (PD-1 inhibitor) •Drug: Venetoclax (Bcl-2 inhibitor)	2020	Suspended
NCT03812796	•Cancer •GI Cancer	•Drug: Domatinostat (HDAC inhibitor) •Drug: Avelumab (PD-1 inhibitor)	2019	Unknown status
NCT03765229	•Melanoma	•Drug: Entinostat (HDAC inhibitor) •Drug: Pembrolizumab (PD-1 inhibitor)	2019	Recruiting
NCT03854474	•Locally Advanced Urothelial Carcinoma •Metastatic Urothelial Carcinoma •Stage III Bladder Cancer AJCC v8 •Stage IIIA Bladder Cancer AJCC v8 •Stage IIIB Bladder Cancer AJCC v8 •Stage IV Bladder Cancer AJCC v8 •Stage IVA Bladder Cancer AJCC v8 •Stage IVB Bladder Cancer AJCC v8	•Biological: Pembrolizumab (PD-1 inhibitor) •Drug: Tazemetostat (EZH2 inhibitor)	2019	Recruiting
NCT03233724	•Non-Small-Cell Lung Carcinoma •Lung Cancer •Non-Small Cell Lung Cancer •Esophageal Carcinoma •Malignant Pleural Mesotheliomas	•Drug: Decitabine (DNMT inhibitor) •Drug: Tetrahydrouridine (inhibitor of cytidine deaminase) •Drug: Pembrolizumab (PD-1 inhibitor)	2018	Recruiting
NCT03445858	•Childhood Solid Tumor •Childhood Lymphoma •Relapsed Cancer •Refractory Cancer •Adult Solid Tumor •Adult Lymphoma	•Drug: Pembrolizumab (PD-1 inhibitor) •Drug: Decitabine (DNMT inhibitor)	2018	Active, not recruiting
NCT03161223	•T-Cell Lymphoma	•Drug: Durvalumab (PD-L1 inhibitor) •Drug: Pralatrexate (dihydrofolate reductase inhibitor) •Drug: Romidepsin (HDAC inhibitor) •Drug: 5-Azacitidine (Methyltransferase inhibitor)	2018	Recruiting
NCT02664181	•Lung Cancer •Non-Small Cell Lung Cancer	•Drug: Nivolumab (PD-1 inhibitor) •Drug: Oral decitabine (DNMT inhibitor) •Drug: Tetrahydrouridine (inhibitor of cytidine deaminase)	2017	Active, not recruiting
NCT03206047	•Platinum-Resistant Fallopian Tube Carcinoma •Platinum-Resistant Ovarian Carcinoma •Platinum-Resistant Primary Peritoneal Carcinoma •Recurrent Fallopian Tube Carcinoma •Recurrent Ovarian Carcinoma •Recurrent Primary Peritoneal Carcinoma	•Drug: Atezolizumab (PD-L1 inhibitor) •Biological: DEC-205/NYESO-1 Fusion Protein CDX-1401 (vaccine that may help the immune system specifically target and kill cancer cells) •Drug: Guadecitabine (DNMT inhibitor) •Drug: Poly ICLC (induces immunohematopoietic cells)	2017	Active, not recruiting
NCT03250273	•Metastatic Cholangiocarcinoma •Cholangiocarcinoma •Pancreatic Cancer •Metastatic Pancreatic Cancer •Unresectable Pancreatic Cancer •Unresectable Cholangiocarcinoma	•Drug: Entinostat (HDAC inhibitor) •Drug: Nivolumab (PD-1 inhibitor)	2017	Completed
NCT02915523	•Epithelial Ovarian Cancer •Peritoneal Cancer •Fallopian Tube Cancer	•Drug: Entinostat (HDAC inhibitor) •Drug: Avelumab (PD-1 inhibitor)	2017	Unknown status
NCT03024437	•Metastatic Cancer •Renal Cancer	•Drug: Atezolizumab (PD-L1 inhibitor) •Drug: Bevacizumab (VEGF inhibitor) •Drug: Entinostat (HDAC inhibitor)	2017	Suspended
NCT02437136	Non-Small Cell Lung Cancer •Melanoma •Mismatch Repair-Proficient Colorectal Cancer	•Drug: Entinostat (HDAC inhibitor) •Drug: Pembrolizumab (PD-1 inhibitor)	2017	Active, not recruiting
NCT02959437	•Solid Tumors •Advanced Malignancies •Metastatic Cancer	•Drug: Azacitidine (DNMT inhibitor) •Drug: Pembrolizumab (PD-1 inhibitor) •Drug: Epacadostat (indoleamine2,3-dioxygenase inhibitor) •Drug: INCB057643 (BET inhibitor) •Drug: INCB059872 (LSD1 inhibitor)	2017	Terminated
NCT02816021	•Melanoma and Other Malignant Neoplasms of Skin •Metastatic Melanoma	•Drug: Azacitidine (DNMT inhibitor) •Drug: Pembrolizumab (PD-1 inhibitor)	2017	Active, not recruiting
NCT02890329	•Previously Treated Myelodysplastic Syndrome •Recurrent Acute Myeloid Leukemia •Recurrent Acute Myeloid Leukemia with Myelodysplasia Related Changes •Recurrent Myelodysplastic Syndrome •Refractory Acute Myeloid Leukemia •Refractory Myelodysplastic Syndrome •Secondary Acute Myeloid Leukemia •Secondary Myelodysplastic Syndrome	•Drug: Decitabine (DNMT inhibitor) •Biological: Ipilimumab (CTLA-4 antibody)	2017	Active, not recruiting
NCT03019003	•Head and Neck Cancer	•Drug: Oral Decitabine (DNMT inhibitor) •Drug: Durvalumab (PD-L1 inhibitor)	2017	Active, not recruiting
NCT03066648	•Leukemia •Myeloid Leukemia •Acute Myeloid Leukemia •Myelodysplastic Syndromes •Preleukemia •Bone Marrow Diseases •Hematologic Diseases •Chronic Myelomonocytic Leukemia	•Drug: Decitabine (DNMT inhibitor) •Drug: PDR001 (PD-1 antibody) •Drug: MBG453 (Tim3 antibody) •Drug: Azacitidine (DNMT inhibitor)	2017	Active, not recruiting
NCT02951156	•Diffuse Large B-cell Lymphoma	•Biological: Avelumab (PD-1 inhibitor) •Biological: Utomilumab (binds to CD-137 protein receptor and stimulates/increases the number of immune cells) •Biological: Rituximab (chimeric monoclonal antibody against the protein CD20) •Other: Azacitidine (DNMT inhibitor) •Drug: Bendamustine (chemotherapy medication) •Drug: Gemcitabine (induces interferon signaling) •Drug: Oxaliplatin (inhibits the synthesis of deoxyribonucleic acid (DNA))	2016	Terminated
NCT02900560	•Epithelial Ovarian Cancer	•Drug: CC-486 (hypomethylation of DNA) •Biological: Pembrolizumab (PD-1 inhibitor)	2016	Terminated
NCT02512172	•Colorectal Cancer	•Drug: Oral CC-486 (hypomethylation of DNA) •Drug: Romidepsin (HDAC inhibitor) •Drug: MK-3475 (PD-1 inhibitor)	2016	Completed
NCT02395627	Breast Neoplasms	•Drug: Tamoxifen (antiestrogens) •Drug: Vorinostat (HDAC inhibitor) •Drug: Pembrolizumab (PD-1 inhibitor)	2015	Terminated
NCT02608437	•Metastatic Melanoma	•Drug: SGI-110 (DNA methylation inhibitor) •Drug: Ipilimumab (CTLA-4 antibody)	2015	Unknown status
NCT02453620	•Breast Adenocarcinoma •Invasive Breast Carcinoma •Metastatic Breast Carcinoma •Metastatic Malignant Solid Neoplasm •Stage III Breast Cancer AJCC v7 •Stage IIIA Breast Cancer AJCC v7 •Stage IIIB Breast Cancer AJCC v7 •Stage IIIC Breast Cancer AJCC v7 •Stage IV Breast Cancer AJCC v6 and v7 •Unresectable Solid Neoplasm	•Drug: Entinostat (HDAC inhibitor) •Biological: Ipilimumab (CTLA-4 antibody) •Biological: Nivolumab (PD-1 inhibitor)	2015	Active, not recruiting
NCT02546986	•Non-Small-Cell Lung Carcinoma	•Drug: CC-486 (hypomethylation of DNA) •Drug: Pembrolizumab (PD-1 inhibitor)	2015	Active, not recruiting
NCT02397720	•Acute Bilineal Leukemia •Acute Biphenotypic Leukemia •Acute Myeloid Leukemia Arising from Previous Myelodysplastic Syndrome •Chronic Myelomonocytic Leukemia •Myelodysplastic Syndrome •Recurrent Acute Myeloid Leukemia •Refractory Acute Myeloid Leukemia •Secondary Acute Myeloid Leukemia •Therapy-Related Acute Myeloid Leukemia	•Drug: Azacitidine (DNMT inhibitor) •Biological: Ipilimumab (CTLA-4 antibody) •Biological: Nivolumab (PD-1 inhibitor)	2015	Recruiting
NCT02608268	•Advanced Malignancies	•Drug: MBG453 (Tim3 antibody) •Drug: PDR001 (PD-1 antibody) •Drug: Decitabine (DNMT inhibitor)	2015	Active, not recruiting
NCT01834248	•Acute Myeloid Leukemia •Alkylating Agent-Related Acute Myeloid Leukemia •Chronic Myelomonocytic Leukemia •Myelodysplastic Syndrome •Refractory Anemia with Excess Blasts	•Biological: DEC-205/NYESO-1 Fusion Protein CDX-1401 (vaccine that may help the immune system specifically target and kill cancer cells) •Drug: Decitabine (DNMT inhibitor) •Drug: Poly ICLC (induces immunohematopoietic cells)	2013	Completed
NCT01928576	•Non-Small Cell Lung Cancer •Epigenetic Therapy	•Drug: Azacitidine (DNMT inhibitor) •Drug: Entinostat (HDAC inhibitor) •Drug: Nivolumab (PD-1 inhibitor)	2013	Recruiting

## Data Availability

Not applicable.

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
