# Peer review of "Epigenetic Perspective of Immunotherapy for Cancers"

_cells, 2023, doi:10.3390/cells12030365_

Round 1

Reviewer 1 Report

Thanks for bringing important insights into the epigenetic regulation of immunotherapy for cancers. This is a comprehensive review with several minor points to be addressed before publication.

1. Table 1 should be a Three-line table.

2. Epigenetic Modifiers in T cells should be discussed as an independent paragraph.

3. CAR-T therapy is mentioned in the introduction, but the discussion of its epigenetic modification is missing. This should be clarified. This issue could also be applied to cancer vaccines and immune modulators.

4. Some recently emerging checkpoints are not mentioned. E.g. VISTA, CD38, IDO, CD47, CD93, and CD161.

5. The authors could find relevant citations to immunotherapy. E.g. "DOI: 10.1186/s13045-022-01325-0", "DOI: 10.1186/s12943-022-01669-8"

6. It is recommended to summarize the information in 3.1 Transcription Factor circuitry in ICT resistance in Table.

Author Response

Response to Reviewer Comments (CELLS-2070789)

Thank you for your very thoughtful comments on our manuscript “Epigenetic Perspective of Immunotherapy for Cancers” (CELLS-2070789). We have addressed the reviewers’ comments in detail here, and hope our resubmission is now suitable for publication in CELLS.

Comments and Suggestions for Authors

Thanks for bringing important insights into the epigenetic regulation of immunotherapy for cancers. This is a comprehensive review with several minor points to be addressed before publication.

We sincerely thank the reviewer for his thoughtful input and excellent comments.

  1. Table 1 should be a Three-line table.

Answer: We wish to thank the reviewer for suggestion. We have updated the table in Three-line format and made it more reader friendly.

  1. Epigenetic Modifiers in T cells should be discussed as an independent paragraph.

Answer: We thank the reviewer for valuable suggestion. We have updated the manuscript with independent paragraph highlighting epigenetic modifiers in T cells at Pages#2-3 Lines#87-111.

  1. CAR-T therapy is mentioned in the introduction, but the discussion of its epigenetic modification is missing. This should be clarified. This issue could also be applied to cancer vaccines and immune modulators.

Answer: We thank the reviewer for excellent comment. We have updated the manuscript with independent paragraph highlighting epigenetic modifiers in CAR T cells at Page#9 Lines# 344-368. However, there here has been limited research on the use of epigenetic modifiers in cancer vaccines and immune modulators, and thus, those topics have not been included in the manuscript.

  1. Some recently emerging checkpoints are not mentioned. E.g. VISTA, CD38, IDO, CD47, CD93, and CD161.

Answer: We thank the reviewer for valuable suggestion. We have updated the manuscript with updated list of emerging immune checkpoints like VISTA, CD38, IDO, CD47, CD93, and CD16 at page#3 Lines#121-124.

  1. The authors could find relevant citations to immunotherapy. E.g. "DOI: 10.1186/s13045-022-01325-0", "DOI: 10.1186/s12943-022-01669-8"

Answer: We thank the reviewer for suggestion we have included relevant citations in manuscript including reviewer’s suggested as Reference # 12  for “DOI: 10.1186/s13045-022-01325-0", Reference #17 for "DOI: 10.1186/s12943-022-01669-8

  1. It is recommended to summarize the information in 3.1 Transcription Factor circuitry in ICT resistance in Table.

Answer: We wish to thank the reviewer for suggestion but summarizing in table won’t be appropriate as studies with an epigenetic perspective on specific transcription circuitry in immunotherapy is not well explored yet and therefore would not fit well in the current form of the manuscript.

Reviewer 2 Report

The presented review, titled “Epigenetic perspective of immunotherapy for cancers”, is focused on the analysis of interconnection between the epigenetic remodeling and immunotherapy response in cancer cells. Indeed, the authors explained how epigenetic events, occurring in cancer cells as well as in immune cells of tumor microenvironment, could promote immune escape of neoplastic cells and/or defective activation of host immune system leading to consequently resistance to immunotherapy.

The topic of this manuscript is very interesting, and I appreciated their effort to write a exhaustive review, in a concise manner, with the purpose to offer a starting point for study novel combinatorial treatments based on immunotherapy and epigenetic drugs . However, the paper appears difficult in this form due to the numerous knowledge (epigenetics, immunotherapy, cancer immunity, immune activation/suppression, etc.) the readers should have for a deep comprehension of the review. In my opinion, the authors took the basilar knowledge of these topics excessively for granted making some passages of the text difficult to understand. I suggest the authors to explain shortly what are: i) epigenetic silencing/activation by methylation and acetylation (which act “inversely”); ii) immune checkpoint and the drugs in actually use for cancer treatment against this target; iii) immune cells (Tregs, TAMs, NKs, CAFs) involved in immunoresponse/immunoescape of cancer cells; iv) cytockines/chemokines with pro-tumorigenic role. Maybe a summarizing picture of immune system mechanisms, involved in anticancer surveillance, would be suggested.

Some minor revisions are recommended for the authors:

•          to correct passive form of the verb (e.g. line 149 -150“…T cell recognition of tumor antigen based on….; line 310-311 “… the activation of STAT4 required for the development….”) and other verb tenses (e.g. line 31 simple past is uncorrected);

•          to clarify some abbreviations such as TME, ICT/ICTs, TSA;

•          to correct the reference 66 because it is not referred to melanoma. Indeed, the authors could cite the recent scientific work of Niebel et al. published on Clinical Epigenetics 14 (2022) (doi: 10.1186/s13148-022-01270-2) in its place;

•          the authors inverted the reference 102 with reference 103, and vice versa, in the text. Please correct the list of references;

•          please, could authors explicate in which type of tumor demethylation of CXCL4 and hypermethylation of CXCL12 and ESR1 are prognostic factors?

•          what is PDCD1 in line 301? Reading the reference 114, I understood that NR4A3 binding on LEF1 promoter leads to epigenetically upregulation of LEF1 expression, isn’t it?

•          for a better examination and comprehension of Table 1, in my opinion it should be more desirable to organize the table in horizontal orientation.

Beyond these my considerations, I recommend this review to publish on Cells after some important revisions.

Author Response

Response to Reviewer Comments (CELLS-2070789)

Thank you for your very thoughtful comments on our manuscript “Epigenetic Perspective of Immunotherapy for Cancers” (CELLS-2070789). We have addressed the reviewers’ comments in detail here, and hope our resubmission is now suitable for publication in CELLS.

Comments and Suggestions for Authors

The presented review, titled “Epigenetic perspective of immunotherapy for cancers”, is focused on the analysis of interconnection between the epigenetic remodeling and immunotherapy response in cancer cells. Indeed, the authors explained how epigenetic events, occurring in cancer cells as well as in immune cells of tumor microenvironment, could promote immune escape of neoplastic cells and/or defective activation of host immune system leading to consequently resistance to immunotherapy.

The topic of this manuscript is very interesting, and I appreciated their effort to write a exhaustive review, in a concise manner, with the purpose to offer a starting point for study novel combinatorial treatments based on immunotherapy and epigenetic drugs . However, the paper appears difficult in this form due to the numerous knowledge (epigenetics, immunotherapy, cancer immunity, immune activation/suppression, etc.) the readers should have for a deep comprehension of the review. In my opinion, the authors took the basilar knowledge of these topics excessively for granted making some passages of the text difficult to understand. I suggest the authors to explain shortly what are: i) epigenetic silencing/activation by methylation and acetylation (which act “inversely”); ii) immune checkpoint and the drugs in actually use for cancer treatment against this target; iii) immune cells (Tregs, TAMs, NKs, CAFs) involved in immunoresponse/immunoescape of cancer cells; iv) cytockines/chemokines with pro-tumorigenic role. Maybe a summarizing picture of immune system mechanisms, involved in anticancer surveillance, would be suggested.

Answer: We sincerely thank the reviewer for his thoughtful input and excellent comments. We have incorporated changes to make it easier to assimilate the knowledge reviewed in the manuscript. We have incorporated a paragraph at Page#2,Lines#73-79 to briefly explain epigenetic silencing/activation and their effects in cancer development. Immune checkpoints have already been described in Figures1 and 2. Various cell types in tumor immune microenvironment (TIME) have been explained briefly one by one in Figure 2 legend at Pages#7-8, Lines# 277-289. Figure 2 is already providing detailed picture of epigenetic drugs, ICT drugs in TIME.

Some minor revisions are recommended for the authors:

  1. to correct passive form of the verb (e.g. line 149 -150“…T cell recognition of tumor antigen based on….; line 310-311 “… the activation of STAT4 required for the development….”) and other verb tenses (e.g. line 31 simple past is uncorrected);

Answer: We thank the reviewer for suggestion. We have examined whole manuscript carefully and made changes where necessary.

  1. to clarify some abbreviations such as TME, ICT/ICTs, TSA;

Answer: We thank the reviewer for suggestion. We have examined the whole manuscript carefully and made changes where necessary.

  1. to correct the reference 66 because it is not referred to melanoma. Indeed, the authors could cite the recent scientific work of Niebel et al. published on Clinical Epigenetics 14 (2022) (doi: 10.1186/s13148-022-01270-2) in its place;

Answer: We thank the reviewer for valuable suggestion and correction. We have corrected mentioned reference with more appropriate Niebel et al. study “ doi: 10.1186/s13148-022-01270-2” at Page#21,Reference 79 -Line# 617 inserted in main text at Page#4,Line#182.

  1. the authors inverted the reference 102 with reference 103, and vice versa, in the text. Please correct the list of references;

Answer: We thank the reviewer for highlighting the error, we have corrected this error in manuscript at Page#8, Line#312 and Line#313

  1. please, could authors explicate in which type of tumor demethylation of CXCL4 and hypermethylation of CXCL12 and ESR1 are prognostic factors?

Answer: We thank reviewer for comment, demethylation of CXCR4 and hypermethylation of CXCL12 and ESR1 are predictive marker in Breast cancer. We have updated same in the manuscript at Page#8 Line#317.

  1. what is PDCD1 in line 301? Reading the reference 114, I understood that NR4A3 binding on LEF1 promoter leads to epigenetically upregulation of LEF1 expression, isn’t it?

Answer: We thank reviewer for crucial error, we have corrected the error at Page#9 Line#376-379. It now reads as “High expression of NR4A1, NR4A2, and NR4A3 has been related with poor prognosis of many cancers, there are reports which suggest that their binding to the target LEF1 promoters can be regulated through DNA methylation and histone acetylation levels.”

  1. for a better examination and comprehension of Table 1, in my opinion it should be more desirable to organize the table in horizontal orientation.

Answer: We wish to thank the reviewer for suggestion. We have updated the table in Three-line format and made it more reader friendly.

Round 2

Reviewer 2 Report

I appreciated the kind reply of authors and the modifications made to the manuscript in line of my suggestions. I confirm the excellent work of authors in writing this review and I suggest to Editor the acceptance and publishing of this manuscript in this form.